# Explainability of Vision Transformers:
# A Comprehensive Review and New Perspectives

## Abstract

Transformers have had a significant impact on natural language processing and have recently demonstrated their potential in computer vision. They have shown promising results over convolution neural networks in fundamental computer vision tasks. However, the scientific community has not fully grasped the inner workings of vision transformers, nor the basis for their decision-making, which underscores the importance of explainability methods. Understanding how these models arrive at their decisions not only improves their performance but also builds trust in AI systems. This study explores different explainability methods proposed for vision transformers and presents a taxonomy for organizing them according to their motivations, structures, and application scenarios. In addition, it provides a comprehensive review of evaluation criteria that can be used for comparing explanation results, as well as explainability tools and frameworks. Finally, the paper highlights essential but unexplored aspects that can enhance the explainability of vision transformers, and promising research directions are suggested for future investment.
**Keywords:** Explainability, Vision Transformer, ViT, Attention, Pruning.

## 1 Introduction

Artificial Intelligence (AI) has seen remarkable progress in recent years, particularly with the success of Deep Neural Networks (DNNs) in diverse areas such as medical diagnosis, financial applications, risk assessments, and generating images and videos. Despite these accomplishments, the application of DNNs in real-world scenarios is still hindered by their opaque nature. Due to the lack of transparency in DNN decision-making, it is difficult to trust them, and their decisions may contain bias. Such problems have led to the emergence of eXplainable AI (XAI), which aims to make intelligent systems more transparent and understandable (Samek et al., 2019).

Explainability refers to the ability of a system to explain why it arrived at a particular decision or recommendation and describe it in a human-understandable way. With explainability, developers, stakeholders, and end-users can trace back the decision-making process of an AI system and ensure that it aligns with ethical and legal standards. Additionally, it helps to uncover biases, errors, or limitations in the AI models and can consequently inform how to improve the system's performance. End-users are more likely to trust an AI system if they understand why it makes a particular suggestion or prediction (Arrieta et al., 2020).

The explanations are even more important in safety-critical fields like healthcare, self-driving cars, finance, or criminal justice, where the models' results may directly impact people's mental or physical health. For instance, consider a medical setting where an AI model is set up to predict a tumor for a given image. In this situation, users may be interested in understanding false positive/negative results, in which healthy cases are labeled as malignant and malignant instances are labeled as healthy cases (Tjoa & Guan, 2020). This feature can be achieved by applying different vision-based explainability methods to the target image and it empowers users to pinpoint which parts of the images misled the models into making such wrong predictions. In addition, from the perspective of the developers, explaining complex models and wrong decisions facilitates the design, development, and debugging process of the models (Ghorbani et al., 2019).

Explainability methods typically follow several approaches with respect to the scope of the methods. They include local and global approaches (Samek et al., 2021), where local methods aim to explain the reasoning behind specific decisions and global methods attempt to provide a broader understanding of the models. In the context of vision-based applications, various types of these methods such as saliency maps (Simonyan et al., 2014), gradient-based methods (Selvaraju et al., 2016; Ancona et al., 2017), perturbation-based methods (Fong & Vedaldi, 2017), and feature visualization (Nguyen et al., 2019) have been studied in the literature. However, the majority of these studies conclude by merely explaining Convolution Neural Networks (CNNs) as a dominant paradigm (Haar et al., 2023), while of the various network architectures, vision transformers, inspired by the great success of self-attention mechanism for language-based tasks, have recently shifted into focus.

Transformers and attention-based models have revolutionized the area of Natural Language Processing (NLP) (Vaswani et al., 2017). They empower language models to process large amounts of data and provide state-of-the-art results in various NLP tasks. In traditional NLP models, such as recurrent neural networks or CNNs, long-term dependencies between words were difficult to capture, leading to a limited capability to process long sequences of text data. In contrast, the transformer-based models use a self-attention mechanism to pay attention to different parts of an input sequence based on its context. It enables the model to capture long-term dependencies and process large inputs. Attention-based models were also introduced to allow neural networks to selectively focus on specific parts of input data relevant to the given task. This mechanism improved the models' performance and made them more interpretable, as the attention scores can be used to identify which parts of the input data influenced the model's decision-making process (Niu et al., 2021).

The success of transformer-based models in NLP has inspired the development of a new class of models in the field of Computer Vision (CV) known as Vision Transformers (ViT) (Dosovitskiy et al., 2020). These models use the architecture of the transformers to process image data, making them suitable for image recognition, object detection, and other CV tasks that are typically solved with CNNs (Touvron et al., 2021b). ViT models use pre-training on large datasets of images to learn visual features, and then, the pre-trained models are fine-tuned on specific tasks for better performance.

Visual transformers are a significant breakthrough in the application of CV as they offer comparable results with CNNs (Liu et al., 2023). Cordonnier et al. (2020) conducted theoretical research to study the equivalence between multi-head self-attention and CNNs. They used patch downsampling and quadratic position encoding to verify their findings. Later, building on this work, Dosovitskiy et al. (2020) applied pure transformers for large-scale pre-training and SoTA performance over many benchmarks. Visual transformers have also delivered exceptional results in other CV tasks including detection (Carion et al., 2020), segmentation (Wang et al., 2021; Cheng et al., 2021), and tracking (Chen et al., 2021). One impressive feat of transformers is generation tasks using multi-modal Large Language Models (LLMs) (Jiang et al., 2021a; Koh et al., 2023; Lee et al., 2023), which are able to generate vision-based content that combines multiple forms of data, such as text and images.

To advance our understanding of vision transformers, it is crucial to explore their inner working procedure and examine their explainability. Among different explainability techniques, some approaches concentrate on developing models that are able to explain themselves inherently. These self-explaining methods may attempt to take advantage of different intrinsic explainable tools such as decision trees or rule-based systems, or from another perspective, they may enforce sparsity in the model architecture. However, these models often struggle to achieve the same level of accuracy as more complex black-box models, and as a result, there must be considered a careful balance between explainability and performance (Rudin, 2019).

In contrast to the inherently/intrinsic explainable methods, interpretable methods attempt to provide human-understandable representations of the decision-making process of the models using different strategies. These methods aim to translate complex, high-dimensional representations into a simpler, more comprehensible format. Explainability methods as debugging tools, gaining insights into the limitations, unintended biases, risks, and social impacts of the models. This enables us to develop reliable, ethical, and safe models that can effectively be deployed in real-world scenarios. Additionally, they facilitate tracking of the models' capabilities over time, compare them with other models, and consequently enhance the model's performance (Ali et al., 2023).

To achieve the above goals, we conducted a comprehensive review of explainability techniques tailored to vision transformers. Our paper presents a systematic categorization of the existing methods, with detailed descriptions and analyses for the more representative ones. We also touch on related works to provide a complete overview. Future research directions are also highlighted for further exploration and development in this area.

The rest of this paper is organized as follows: section 2 introduces the fundamental background, concepts, and notation. In section 3, a comprehensive taxonomy for the explainability of vision transformers is presented and elaborated upon. Evaluative criteria, tools, and frameworks, as well as datasets specific to the XAI literature, are covered in sections 4, 5, and 6, respectively. Section 7 provides the results of experiments conducted on several well-known methods previously introduced. Section 8 delves into the research challenges of this field and outlines the potential future works. Lastly, section 9 summarizes our findings and contributions.

## 2 Preliminaries

### 2.1 Background of the transformers

Transformers revolutionized machine translation by overcoming the constraints of sequential transduction models. Those models that rely on an encoder-decoder setup with a fixed-size context vector and struggle to maintain information over long sequences. Indeed, Transformers eliminated recurrent neural networks within blocks, improving parallel processing and mitigating challenges like vanishing and exploding gradients (Vaswani et al., 2017; Sutskever et al., 2014).

The transformer's architecture typically contains a positional encoding block, self-attention heads, and feed-forward layers. The encoding block adds positional information into the input sequence to understand the distance between different words of the sequence. The attention heads assign the importance of different parts of the input sequence with respect to each other, and consequently are able to accurately capture the complex relationships between them. Finally, the feed-forward layers receive the results of the multi-head self-attention components and process them to generate the output (Devlin et al., 2018).

### 2.2 What is self-attention?

The key component in the transformer's architecture is the self-attention layer, which attempts to understand the relationships/dependencies of different elements of a sequence. Indeed, the weight of each element is determined by its similarity to the other elements in the sequence, and therefore, the attention scores of the token pairs are computed. In the NLP applications, the tokens commonly correspond to different words or different parts of a sentence, while in the vision context, the tokens are associated with different patches of an image.

The self-attention mechanism is applied in the self-attention layers of the transformers. This mechanism involves computation of the scaled dot-product attention using three vectors of query($Q$), key($K$), and value($V$) as follows:

$$\text{Attention}(Q, K, V) = \text{Softmax}\left(\frac{QK^T}{\sqrt{d_k}}\right) V \qquad (1)$$

where $d_k$ is the dimensionality of the key vectors. $Q$, $K$, and $V$ matrices are obtained using the input values of each layer and the parameter matrices $W$ as follows:

$$Q = QW^Q, \quad K = KW^K, \quad V = VW^V \qquad (2)$$

in which queries, keys, and values are initiated with the input vector and subsequently come from the output of the previous layer.

Since a single attention head can only focus on one feature at a time, the Multi-Head Self-Attention mechanism (MHSA) was introduced. This mechanism enriches the diversity of the features without extra costs. It involves processing the input features using several parallel heads independently. The resulting vectors

are combined and transformed to produce the final outputs. As these statements, the attention scores are calculated using $H$ heads as follows:

$$Z_h = \text{Attention}(Q_h, K_h, V_h), \quad h = 1, \cdots, H \tag{3}$$

where the heads' results are concatenated and projected through a feed-forward layer with the parameter matrix $W^O$ as follows:

$$\text{MultiHead}(Q, K, V) = \text{Concat}(Z_1, Z_2, \cdots, Z_H) \, W^O \tag{4}$$

In the language-based transformers, the multi-head self-attention mechanism is employed in three different ways, namely encoder self-attention, decoder self-attention, and encoder-decoder self-attention, while in the vision-based applications, they are primarily utilized in the encoder blocks.

### 2.3 Vision transformers

Following the developments of the transformers in NLP, different works attempted to take advantage of them in visual applications, and consequently, ViTs were introduced by (Dosovitskiy et al., 2020). The transformer blocks in ViTs allow global integration of information across the entire image using the multi-head self-attention mechanism. They represent an image as a sequence of tokens. They indeed consist of a patch embedding layer that partitions each image into $N \times N$ patches. The obtained 2D patches are then flattened into 1D representations. They are treated as tokens and transformed into feature vectors within the transformer framework.

To perform the classification task, a learnable classification token is added to the image representations. Similar to the base Transformer's architecture, position embedding is added to each patch to retain positional information. ViT networks consist of some other components such as a multi-layer perceptron with Gaussian Error Linear Unit (GELU) activation, a layer normalization before each block to enhance training and generalization, and a Residual connection to facilitate direct gradient flow through the network. An illustration of the ViT architecture is found in Fig. 1.

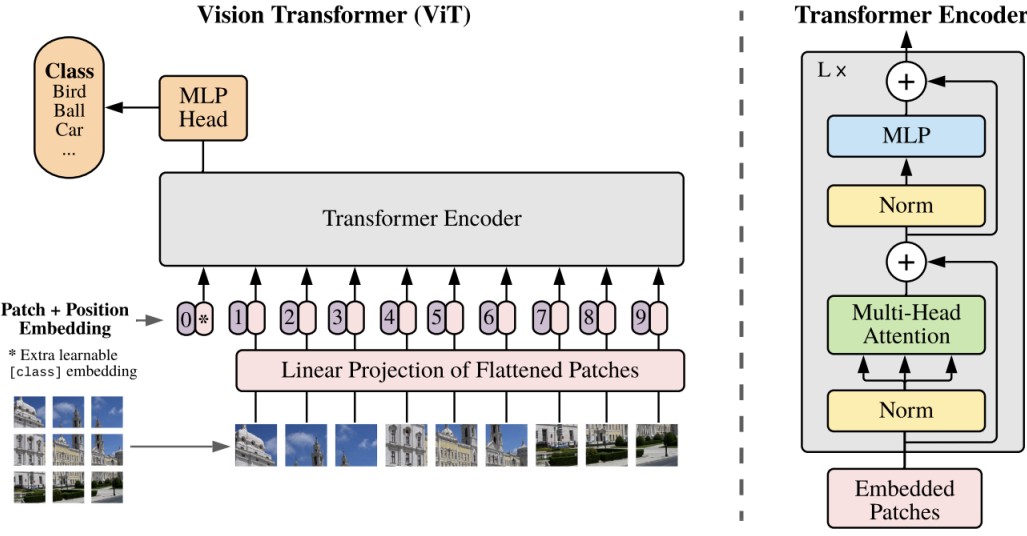

Figure 1: The typical architecture of Vision Transformer (ViT) (Dosovitskiy et al., 2020).

# 3 Explanation of vision transformers

Following the breakthrough works presented based on vision transformers for various CV domains, multiple approaches have emerged to enhance the explainability of these networks. However, a comprehensive survey is necessary to gain a better understanding of these methods and to identify potential areas for improvement. With a focus on classification task[1], this section presents an overview of existing explainability techniques for visual transformers. To provide a clear overview, we categorize and summarize these methods into five distinct groups based on their working procedures, motivations, and structural characteristics, as illustrated in Fig. 2.

In the following, the most prominent works from each group are discussed in detail. Alongside this, a brief introduction of other relevant works is provided. We start with the common attribution methods that are generally applied/adapted for explaining vision-based architectures. Then, We move on to the methods that specifically visualize or do an operation on the attention weights. Additionally, we introduce the pruning-based methods, which implicitly or explicitly influence the explainability of vision transformers. Finally, a brief discussion on the inherently explainable methods as well as the methods proposed for the explanation of the other tasks is presented.

## 3.1 Common attribution methods

Feature attribution methods aim to explain how the input features contribute to the output of a machine learning model. These methods particularly measure the relevance of each input feature to the model's prediction. In this context, several approaches including Gradient-Based, Perturbation-based, and Decomposition-Based methods are discussed in the literature. Among these approaches, some methods like GradCAM (Selvaraju et al., 2017) and Integrated Gradients(IG) (Sundararajan et al., 2017) have directly been applied to visual transformers (Böhle et al., 2023), while some others like SHAP (Lundberg & Lee, 2017) and Layer-Wise Relevance Propagation (LRP) (Bach et al., 2015) have been adapted to be employed by the ViT-based architecture.

SHAP method, which obtains pixels' contribution using Shapley values, offers a theoretically sound alternative; however, its computational cost makes it

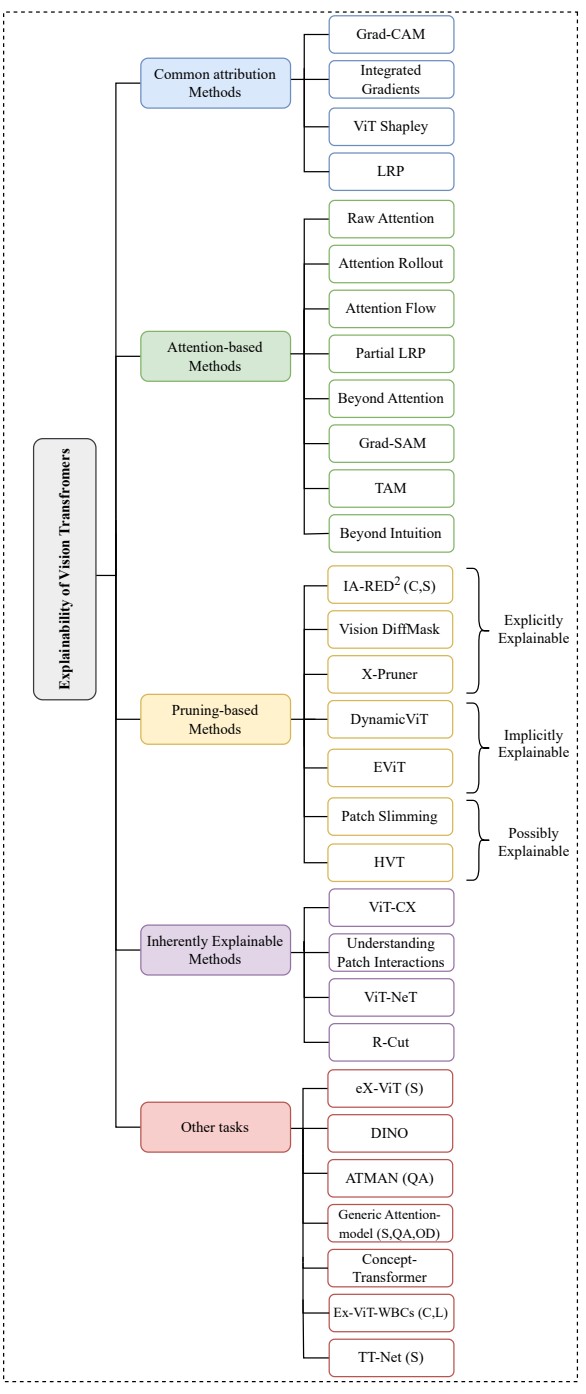

Figure 2: Taxonomy of explainability methods for ViT. The primary task of the methods is classification (not mentioned), with additional tasks specified as follows: Classification(C), Localization(L), Segmentation(S), Question-Answering(QA), and Object Detection(OD).

---

[1]It is straightforward to add layers above the main encoder of the ViT backbone, allowing it to be extended for other downstream tasks.

challenging when dealing with large high-dimensional models.

A recent study, known as ViT Shapley, made Shapley values practical specifically for ViT applications (Covert et al., 2023). They first leverage an attention masking approach to evaluate ViTs with partial information, and then generate Shapley value explanations using a separate, trained explainer model. The LRP method has also been updated for transformers to determine the importance of different heads in the multi-head attention mechanism (Voita et al., 2019; Chefer et al., 2021a). This method is detailed in the next section.

Given our main emphasis lies in the methods specifically designed for the transformer-based models, we briefly discussed some feature attribution methods adapted for ViT. The exclusive techniques, however, are mainly elaborated upon in the subsequent sections.

### 3.2  Attention-based methods

The attention mechanism is a powerful technique that enables a model to identify and prioritize the most relevant parts of an input sequence. Intuitively, it determines the relative importance of different tokens in the input sequence and ensures that important information from all parts of the input sequence is effectively captured and utilized (Vaswani et al., 2017). This ability can particularly be leveraged to gain insights into the model's predictions. Many existing approaches focus specifically on using attention weights or the knowledge encoded therein to explain the model's behavior. These methods include direct visualization of attention weights or applying various functions to them (Zhao et al., 2023).

In vision-based applications, visualizing **raw attention** weights can be an effective tool for researchers to intuitively identify attention patterns across different heads and layers of the network. Fig 3 depicts the attention matrices across the 12 layers of ViT, where the minimum, maximum, or average of the outputs of different heads is computed in each layer. This figure shows that as the network grows deeper, visualizing raw attention is not sufficient and results in less reliable explanations.

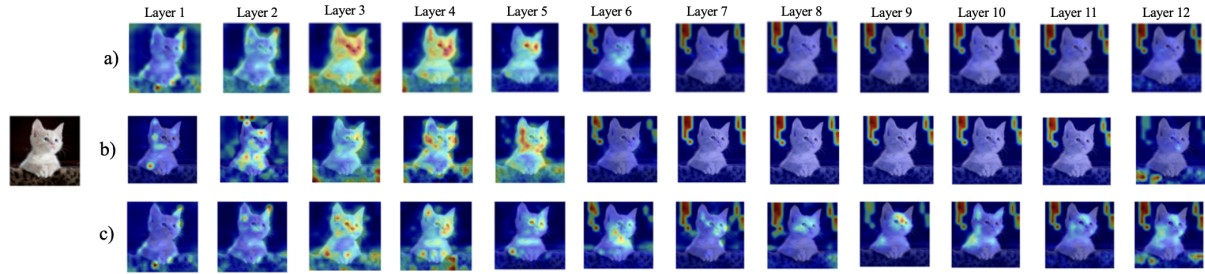

Figure 3: Visualization of the raw attention matrices across the 12 layers of ViT, where a) mean, b) minimum, and c) maximum of the attention heads are computed in each layer.

Indeed, visualizing only the raw attention weights fails to capture the global relationship between input tokens. Moreover, as information from different tokens flows through the network layers, it becomes increasingly intertwined, making it difficult to analyze individual attention scores. Therefore, it is essential to incorporate additional contextual information to provide more reliable explanations in deeper networks. To address these challenges, two methods, namely Attention Rollout and Attention Flow, were introduced by Abnar & Zuidema (2020). These approaches quantify the flow of information and approximate attention to input tokens in a more holistic manner. Although they were initially developed for NLP tasks, both methods have since been extended to vision models (Chefer et al., 2021a).

The **Attention Rollout** technique tracks the transmission of information from input tokens to intermediate hidden embeddings by extending attention weights through different layers of the network. It first takes the minimum, maximum, or mean of the attention heads of each block and then multiplies them recursively with

the attention of the preceding blocks. For a network with $B$ multi-head attention blocks and raw attention matrix of $A$, the Attention Rollout is computed as follows (Chefer et al., 2021a):

$$\hat{A}^{(b)} = I + \mathbb{E}_h A^{(b)} \; ; \; b = 1, ..., B \tag{5}$$

$$Rollout = \hat{A}^{(1)} \cdot \; \hat{A}^{(2)} \cdot ... \cdot \hat{A}^{(B)} \tag{6}$$

in which the identity matrix $I$ is taken into account for the skip connections to avoid self-inhibition of the tokens. $\mathbb{E}_h$ represents the mean value across the $h$ heads dimension. The Rollout method is based on simplistic assumptions and tends to highlight irrelevant tokens. Although it outperforms the raw attention explanations, it has several limitations. One of these is the exaggerated differences between attention scores for different tokens, which occur because it combines tokens linearly through layers based on attention weights. This results in a loss of faithfulness in saliency maps. Faithfulness refers to how accurately a saliency map represents the target model. It is measured by observing how quickly the model's score for the target class changes when important pixels are removed or added. This loss of faithfulness affects evaluation metrics such as Deletion Area Under the Curve (AUC) and Insertion AUC (for a detailed description of these criteria please see Section 4) (Komorowski et al., 2023; Bhatt et al., 2020; Petsiuk et al., 2018; Nguyen & Martínez, 2020).

In the next step, **Attention Flow** was proposed. This method views the attention graph as a flow network and computes maximum flow values from hidden embeddings to input tokens using a maximum flow algorithm. It is known to be more correlated with the importance scores of the input tokens than the Rollout method with respect to the ablation method and input gradients, but it is too slow to support large-scale evaluations (Abnar & Zuidema, 2020). These two methods solely rely on pairwise attention scores, and therefore their results remain fixed for a specific input sample. They are indeed independent of the target class and fail to differentiate between positive and negative contributions of the features to the final decision.

In order to effectively take advantage of the MHSA mechanism, it is important to consider the varying importance of each attention head when analyzing the transformer models. This is a factor that previous techniques did not take into account. To address this issue, researchers have developed a modified version of LRP, known as **PartialLRP**, that focuses only on attention head relevance, rather than propagating scores back throughout all layers (Voita et al., 2019).

Building on the previous work, Chefer et al. (2021a) proposed **Beyond Attention** method to compute LRP-based scores for every attention head across all layers ($\bar{A}^{(b)}$). This method integrates both attention head's relevance and its gradient with respect to the input ($\nabla A^{(b)}$) as follows:

$$\bar{A}^{(b)} = I + \mathbb{E}_h (\nabla A^{(b)} \odot R^{(n_b)})^+ \; ; \; b = 1, ..., B \tag{7}$$

$$C = \bar{A}^{(1)} \cdot \; \bar{A}^{(2)} \cdot ... \cdot \bar{A}^{(B)} \tag{8}$$

in which $n_b$ is the layer that corresponds to the softmax operation in block $b$, and $R^{(n_b)}$ is the relevance of this layer with respect to the target class $t$. The Hadamard product $\odot$ and the matrix multiplication $(\cdot)$ are used to compute the scores. To obtain the weighted attention relevance, only the positive values resulting from the gradients-relevance multiplication are taken into account, mimicking the notion of positive relevance. By applying this process, the class-specific visualizations for the self-attention models can be generated. This approach has since been extended to multi-modal transformers by exploring variations of the attention mechanism (beyond self-attention) (Chefer et al., 2021b).

The limitations of relying solely on raw attention matrices to explain model predictions have prompted researchers to apply functions, such as gradients, to the raw attention weights. One such technique is **Grad-SAM**, initially developed for transformers but also adapted for vision applications by Barkan et al. (2021). This involves calculating the partial derivatives of the model output with respect to the attention blocks, denoted by $G_x^{lh}$, as follows:

$$G_x^{lh} := \frac{\nabla s_x}{\nabla A_x^{lh}} \tag{9}$$

in which $s_x$ represents the logit score, and $A_x^{lh}$ refers to the attention matrix of the specified head $h$ in layer $l$ for input $x$. Through these calculations, many activation values are close to zero and have negative gradients. When these negative gradients accumulate, they can overpower the small number of positive gradients that carry vital information. As a solution, the gradients are passed to the ReLU function to zero out the negative values as follows:

$$H_x^{lh} = A_x^{lh} \odot ReLU(G_x^{lh}) \tag{10}$$

Finally, the importance of token $x_i$ $(i = 1, ..., N)$ with respect to prediction $s_x$ is computed by $r_{x_i}$ as:

$$r_{x_i} = \frac{1}{LHN} \sum_{l=1}^{L} \sum_{h=1}^{H} \sum_{j=1}^{N} [H_x^{lh}]_{ij} \tag{11}$$

where the higher values of $r_{x_i}$ indicates the higher importance of $x_i$ in that certain prediction.

Another work, known as Transition Attention Maps (**TAM**) by Yuan et al. (2021), considers the information flow inside ViT as a Markov process. It leverages the hidden states of the tokens to denote the information of each layer, which is recursively evolved through the Transformer's processing procedure. This method propagates information from top to bottom. It computes the relevance between high-level semantics and input features. High-level semantics are used to represent richer patterns and concepts and are processed through the attention mechanism in layers. The input features include edges, textures, and basic shapes. Furthermore, to exhibit the class discriminative ability, TAM assigns the importance scores by incorporating the idea of Integrated Gradient (Sundararajan et al., 2017) and Grad-CAM (Selvaraju et al., 2017).

**Beyond Intuition** (Chen et al., 2023), is also a novel interpretation framework to approximate token contributions. This framework relies on the partial derivative of the loss function to each token and operates in two stages, namely attention perception and reasoning feedback. Attention perception considers the relationship between input and output in each attention block, leading to the derivation of two recurrence formulas with head-wise and token-wise attention maps. Specifically, the token-wise attention map performs 20 times more accurately than Attention Rollout in terms of faithfulness evaluation metrics, such as Deletion AUC, Insertion AUC, and Faithfulness correlation (see section 4) (Komorowski et al., 2023; Bhatt et al., 2020; Petsiuk et al., 2018; Nguyen & Martínez, 2020).

Fig 4 visualizes the results of the attention-based methods for two samples. It is important to note that although concentrating on attention weights is a popular interpretation approach, there is a debate in research regarding their utility for explainable purposes, as some experts believe that they are merely one component in a series of several nonlinear operations and may not be as informative as initially assumed. Therefore, there is a need for further exploration of their effectiveness for explanations. This matter will be covered later in the discussion.

### 3.3 Pruning-based methods

Pruning is a powerful approach that is widely used to optimize the efficiency and complexity of transformers. They attempt to remove redundant or uninformative elements such as tokens, patches, blocks, or attention heads of the networks. These techniques have shown promising results in improving the explainability of the ViT-based architectures as well. In this context, some of the methods are explicitly developed for explainability purposes, while some others focus primarily on improving efficiency and do not specifically target explainability. The former methods focus on pruning some model components specifically to enhance the explainability of vision transformers. On the other hand, the other methods apply pruning techniques to increase efficiency, however, they have been shown to positively impact model explainability as well. These methods do not explicitly prioritize explainability as their primary goal and this is why we call them implicitly explainable methods. Indeed, the explainability improvement is a beneficial byproduct of their design. In addition, there is a third group of pruning methods found in the literature that have not basically

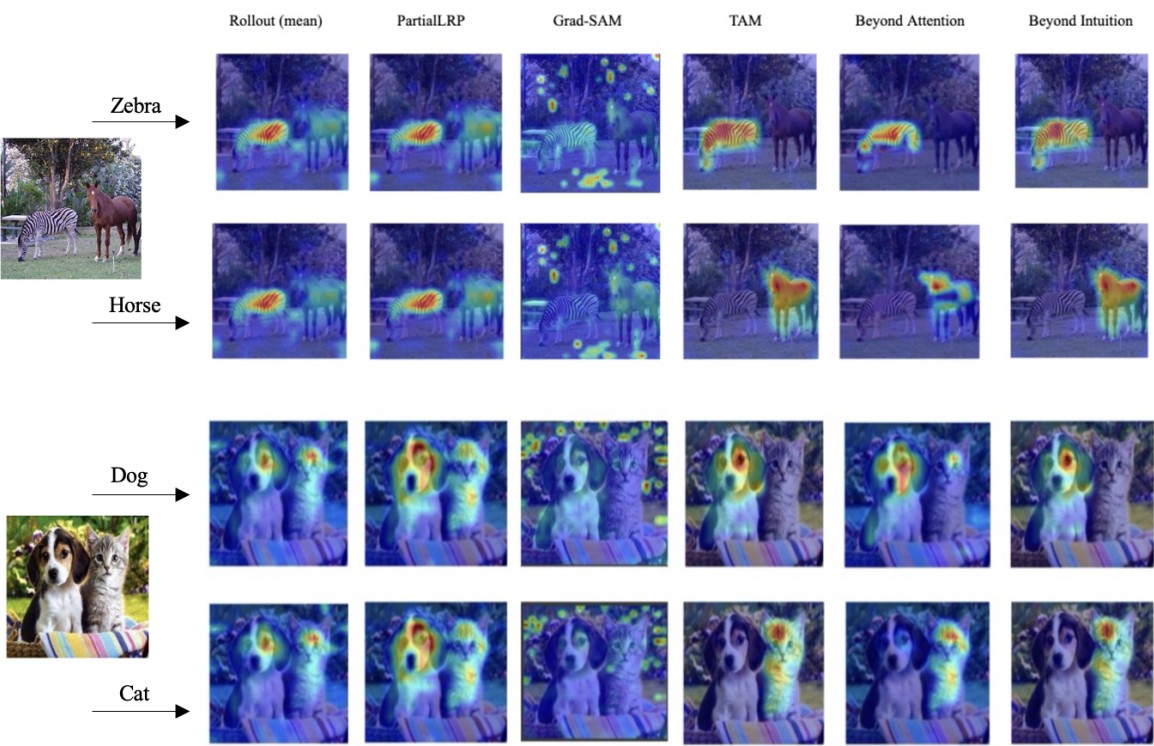

Figure 4: Class-specific visualization of the attention-based methods.

mentioned explainability in their findings, but they may still have favorable implications for explainability. As these statements, we categorize the ViT-based pruning methods into three groups: explicitly explainable, implicitly explainable, and possibly explainable methods, which are elaborated upon below.

### 3.3.1 Explicitly explainable methods

Among the pruning-based methods, several notable approaches exclusively aim to provide less complex and more interpretable models. One such method is **IA-RED²** (Pan et al., 2021a), which strives to find the perfect balance between efficiency and interpretability while maintaining the versatility and flexibility of the original ViT. This method dynamically drops less informative patches, reducing the input sequence length. IA-RED² divides the original ViT structure into distinct components: a multi-head interpreter and processing blocks. The interpreter assesses the informativeness of the patch tokens, discarding those with scores below a certain threshold, and passing the remaining ones through the processing blocks. IA-RED² achieved significant speedups (up to $1.4x$) with minimal loss of accuracy. Furthermore, the authors demonstrated that the interpretable module learned by IA-RED² outperforms existing interpretation methods.

**X-Pruner** (Yu & Xiang, 2023) is a novel method specifically designed to prune less significant units. This method initially creates an explainability-aware mask to measure the contribution of every prunable unit in predicting a particular class. Following this, it removes units with the lowest mask values. X-Pruner accomplished substantial FLOP savings and is able to produce correct, compact, and less noisy visual explanation maps on pruned models.

**Vision DiffMask** (Nalmpantis et al., 2023) is an adaptation of a method originally developed for language processing (De Cao et al., 2020). It aims to identify the minimal subset of an image that is required for ViT to make correct predictions. This pruning method comprises gating mechanisms on each ViT layer, which are optimized to maintain the origin model output when masking input. During training, these gates

vote on whether an image patch should be removed or preserved. The votes are then aggregated to create the final mask. During inference, the gates predict probabilities instead of binary votes, which yields a continuous attribution map over the input. The authors tested Vision DiffMask using faithfulness tasks and perturbation tests. They also showed that Vision DiffMask is capable of clearly delineating the triggering subset from the rest of the image, thus offering a better understanding of the model's predictions.

### 3.3.2 Implicitly explainable methods

Implicitly explainable methods are initially designed to improve model efficiency, but they have clearly demonstrated through visualization that their pruning approaches would result in positive effects on the models' explainability as well. While their main goal is to improve efficiency, the clarity that comes from simplifying the model also provides valuable insights into how the model works and makes decisions. This dual benefit highlights the importance of implicitly explainable methods in enhancing both the efficiency and explainability of vision transformers.

One such approach is the **Dynamic ViT** framework (Rao et al., 2021), which leverages dynamic token pruning to accelerate the ViT operations. The method employs a lightweight prediction module to estimate the importance score of each token based on the current features. This module is added to different layers of ViT to hierarchically prune redundant tokens. Additionally, an attention masking strategy is used to differentiate tokens by blocking their interactions with others. Their experimental results show that the Dynamic ViT model significantly improves the ViT efficiency while maintaining the model's accuracy. Additionally, Dynamic ViT enhances interpretability by locating the critical image parts that contribute most to the classification, step-by-step. Notably, token pruning tends to keep the tokens located in the image center, which is reasonable because objects in most images are located in this area.

Efficient Vision Transformer, **EViT** (Liang et al., 2022), is another innovative approach that aims to speed up ViTs by reorganizing the image tokens. The key concept behind EViT is attentiveness, which assesses the relationship between the image tokens and the class token. Class token is an extra token (CLS) added to the set of image tokens and is responsible for aggregating global image information and final classification. In contrast, image tokens refer to the image patches, which are treated as individual units. By computing the tokens' scores based on attentiveness, EViT retains the most relevant tokens while fusing the less attentive ones into a single token. This reduces the number of tokens and consequently computation costs as the network deepens. EViT not only enhances model accuracy but also maintains similar or smaller computational costs compared to the original DeiT/LV-ViT by Touvron et al. (2021a); Jiang et al. (2021b). Furthermore, authors have demonstrated through visualization that gradually fusing inattentive tokens allows ViT to focus on the class-specific tokens in the images, which leads to better interpretability. In section 7, we conducted several evaluations to compare the results of these two methods in terms of qualitative and quantitative performances.

### 3.3.3 Possibly explainable methods

Moving forward, our attention turned to additional pruning methods that may not have been originally intended to improve the interpretability of ViT but may offer the potential for further research into their impact on the explainability of the models. The first method is **Patch Slimming** (Tang et al., 2022), a novel algorithm that accelerates ViTs by targeting redundant patches in input images using a top-down approach. Specifically, this technique identifies the most informative patches of the last layer and then uses these patches to guide the patch selection process of the previous layers. The ability of this algorithm to selectively retain the key patches has the potential to highlight important visual features, possibly leading to enhanced interpretability. However, despite this potentially compelling feature of the algorithm, the authors did not explicitly discuss it in their research. Patch Slimming has been shown to significantly reduce the ViT computational costs by over 45%, while only resulting in a 0.3% decrease in top-1 accuracy.

Another novel approach is the Hierarchical Visual Transformer, **HVT** (Pan et al., 2021b), which was introduced to enhance the scalability and performance of ViTs. It gradually reduces the sequence length as the model's depth increases. In addition, by partitioning ViT blocks into stages and applying pooling operations at each stage, the computational efficiency is significantly improved. HVT also performs predictions with-

out a class token, which further reduces the computational cost. Empirical results demonstrate that HVT surpasses the DeiT model in various image classification benchmarks while having similar computational expenses. Given the gradual concentration on the most important components of the model, there is an opportunity to explore its potential impact on enhancing explainability and interpretability.

Most of the available pruning methods have been thoroughly evaluated from the efficiency perspective. For instance, Table 1 provides a comparison for some of these pruning methods in terms of their influence on FLOP saving, accuracy drop, and throughput improvement [2]. However, there is a significant gap in the literature regarding the explainability evaluation of these methods. Most of the methods only present visualization and qualitative analysis, without providing quantitative evaluations (see section 4).

Table 1: Comparison the efficiency of different pruning-based methods applied to DeiT-S Touvron et al. (2021a) and ImageNet dataset.

| Pruning Method | GFLOPs ↓ (%) | TOP-1 Accuracy ↓ (%) | Throughput ↑ (%) |
|---|---|---|---|
| IA-RED$^2$ | – | 0.7 | 46 |
| X-Pruner | 47.9 | 1.09 | – |
| DynamicViT | 37 | 0.5 | 54 |
| EViT | 35 | 0.3 | 50 |
| Patch Slimming | 47.8 | 0.3 | 43 |
| HVT | 47.8 | 1.8 | – |

### 3.4 Inherently explainable methods

These explainable methods aim to understand how the model's working procedure, such as the weights and biases of a neural network, contribute to its output. In the context of DNN, they attempt to develop models that are able to explain themselves inherently. They may take advantage of different intrinsic interpretable tools as well.

In this context, **ViT-CX** was introduced by Xie et al. (2022), which is a mask-based explanation method customized for ViT models. This method relies on patch embeddings and their causal impacts on the model output rather than focusing on the attention given to them. The method involves two phases of mask generation and mask aggregation, which consequently results in providing more meaningful saliency maps. The generated explanation by ViT-CX exhibits significantly better faithfulness to the model.

A novel explainable visualization approach was proposed by Ma et al. (2023) to **analyze the patch-wise interactions** in vision transformers. The authors first quantify the patch-wise interaction, and then identify the potentially indiscriminative patches. That is to provide an adaptive attention window, which constrains the receptive field of each patch. According to their observation, the size of the patch's receptive field is correlated to the patch's semantic information, where patches of the primary object of an image tend to have smaller receptive fields compared to those from the background. As a result, local textures or structures are more learned in smaller receptive fields, whereas the correlation between the object and the background is focused in larger receptive fields.

**ViT-NeT** (Kim et al., 2022) introduces a new ViT neural tree decoder that describes the decision-making process through a tree structure and prototype. It enables a visual interpretation of the results as well. This model effectively addresses the classification of fine-grained objects by simultaneously considering the similarities between different classes and variations within the same class.

The **R-Cut** method (Niu et al., 2023) enhanced the explainability of ViTs with two modules, namely the Relationship Weighted Out and the Cut modules. The former module focuses on extracting class-specific information from intermediate layers, emphasizing relevant features. The latter module performs fine-grained feature decomposition, taking factors like position, texture, and color into account. By integrating these modules, dense class-specific visual explainability maps can be generated.

---

[2]All results are based on the original report, if available.

**Ex-ViT-WBCs** (Katar & Yildirim, 2023) is a model designed for the automatic classification and localization of White Blood Cells (WBCs) from blood films. This model uses a self-attention mechanism within the Vision Transformer architecture to effectively extract features from input images. It uses a Multi-Layer Perceptron (MLP) head for classification and incorporates the Score-CAM algorithm to visualize the pixel areas that the model focuses on during its predictions, enhancing its explainability.

Finally, **TT-Net** (Wang et al., 2023) is an explainable ViT used for medical segmentation. It introduces a Tensorized Transformer Network (TT-Net) that employs transformers for accurate segmentation of organs, tissues, and lesions in 3D medical images. The Key innovations of this work include a multi-scale transformer with layer fusion to capture contextual interaction information, a Cross Shared Attention (CSA) module based on perceptual hashing similarity fusion to extract global features, and a Tensorized Self-Attention (TSA) module to reduce the large number of parameters in transformers while being easily embedded into other models. TT-Net provides good explainability by visualizing the attention maps of the transformer layers, allowing users to interpret the model's decision-making process.

### 3.5  Explainability of other tasks

Explainability of ViT-based architecture in other CV tasks beyond classification is still being explored. There are several explainability methods proposed specifically for other tasks and the early results were promising. However, as the field of deep learning continues to advance, we can expect to see more human-understandable applications of ViT to a wide range of image-related tasks. In this section, we overview several state-of-the-art works related to other CV tasks, however, each task may require a thorough investigation on its own in a separate comprehensive study.

**eX-ViT** (Yu et al., 2023) is a novel explainable vision transformer designed for weakly supervised semantic segmentation. It is indeed a Siamese network (Koch et al., 2015) with two branches for processing input pairs (two augmented versions of an original input data) in a self-supervised manner and finally learning interpretable attention maps. Each branch has a transformer encoder with novel explainable multi-head attention and attribute-guided explainer modules. Furthermore, to improve interpretability, an attribute-guided loss module is introduced with three losses: namely global-level attribute-guided loss, local-level attribute discriminability loss, and attribute diversity loss. The former uses attention maps to create interpretable features, while the latter two enhance attribute learning.

In the study presented by Caron et al. (2021), the researchers introduced **DINO** as a simple self-supervised transformer-based method. This method performs as a self-distillation approach with no labels. The authors report that the final learned attention maps are able to effectively preserve the semantic regions of the images, which can further be applied for explanation purposes.

**Generic Attention-model** (Chefer et al., 2021b) was introduced as a novel approach for explaining predictions made by transformer-based architectures. It covers both bi-modal transformers and those with co-attentions. This method was applied to the three most commonly used architectures, namely pure self-attention, self-attention combined with co-attention, and encoder-decoder attention. In order to test the explanation of the models, the authors used the visual question-answering task, however, it is applicable to other CV tasks, such as object detection and image segmentation.

In another work, **ATMAN** (Deb et al., 2023) attempts to understand the transformer predictions through memory-efficient attention manipulation. They present, which is a modality-agnostic perturbation method that leverages attention mechanisms to generate relevance maps of the input with respect to the output prediction.

Rigotti et al. (2022) attempted to generalize attention from low-level input features to high-level concepts. This is achieved through the use of a deep learning module called **Concept-Transformer**. This module generates explanations for the model output by highlighting attention scores over user-defined high-level concepts, ensuring both plausibility and faithfulness.

# 4 Evaluative criteria

In earlier sections, we introduced various explanation techniques specifically developed for ViT-based applications. However, assessing how well these methods depict the model's reasoning process poses different challenges. To address this concern, the literature proposes a range of evaluative criteria, which assist in selecting and designing the most suitable explainability technique. Fig. 5 summarizes these criteria, and we will delve deeper into them in the following.

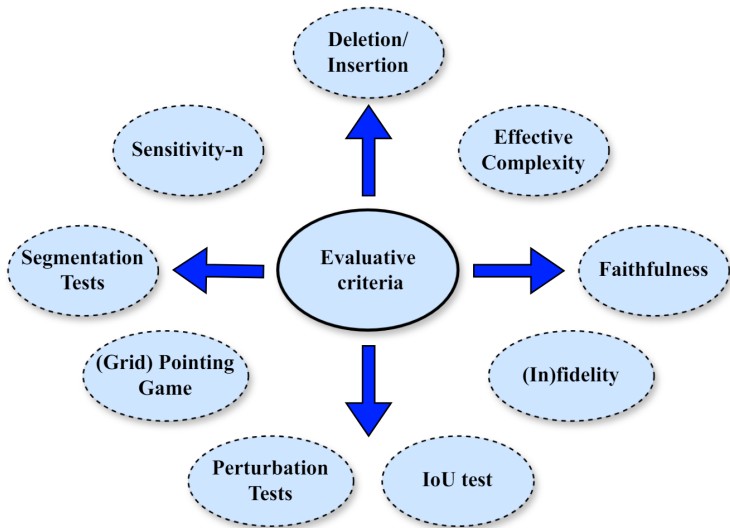

Figure 5: Different criteria for evaluating explainability methods in vision-based applications

• **Deletion and Insertion** are used to evaluate the faithfulness of a saliency map to the target model. They measure how well the saliency map identifies the pixels that are most influential to the model's prediction (Petsiuk et al., 2018). Deletion Area Under the Curve (AUC) calculates how quickly the model's score for the target class declines when the important pixels are omitted from the image. On the other hand, Insertion AUC measures how quickly the model's score for the target class increases when the important pixels are incorporated into a blank canvas to amplify saliency. A smaller Deletion AUC and a larger Insertion AUC indicate that the saliency map accurately identifies the crucial pixels for the model's prediction, and therefore demonstrate higher faithfulness of the saliency map (Xie et al., 2022).

• **Effective Complexity** assesses the number of attributions surpassing a threshold, indicating the importance or insignificance of the corresponding features. This measure is crucial in visualization-oriented explanations. A low effective complexity means that some of the features can be ignored even though they do have an effect because their effect is actually small. While this may lead to a loss of factual accuracy, it can provide a more integrated and simplified explanation (Nguyen & Martínez, 2020).

• **Faithfulness** is a method used to assess the quality of feature attributions without human intervention. It measures how accurately feature attributions align/correlate with a model's predictions. It removes specific features and analyzes the linear correlation between the model's predicted logits and the average explanation attribution corresponding to the subset of features. This process takes multiple runs into account and test samples, which generate a numerical value between -1 and 1 for each input-attribution pair (Komorowski et al., 2023; Bhatt et al., 2020).

• **(In)fidelity** is used to evaluate how well an explanation captures changes in a model's predictions when the input undergoes significant perturbations. The infidelity measure is defined as the expected difference between two terms: the dot product of the perturbation to the explanation and the output perturbation, which represents the difference in function values after significant input changes. The goal is to find an explanation that minimizes this infidelity measure, indicating a high fidelity explanation that accurately

reflects the model's behavior under different perturbation scenarios. Infidelity is a complementary metric to sensitivity, which measures how much an explanation changes when the input data is changed slightly (Yeh et al., 2019).

- **Intersection over Union (IoU) test** is a standard metric for evaluating the performance of object detectors and trackers (Rezatofighi et al., 2019). It has also been used to evaluate the performance of explainability methods by measuring how well the predicted explainability maps overlap with the ground truth bounding boxes of the objects of interest (Niu et al., 2023). In this procedure, first, the explainability feature map is increased in resolution to match the size of the original image so that the predicted explainability bounding box can be more accurately compared to the ground truth bounding box. Then, a threshold is applied to discard background regions. Finally, the IoU score is calculated as the intersection of the predicted and ground truth bounding boxes divided by their union. A higher IoU score indicates that the explainability feature map is better at localizing the object of interest.

- **Perturbation Tests** work by gradually masking out input pixels/tokens based on the explanations provided by the given explainability method. The accuracy of these masked inputs is quantified by the AUC of their accuracies. There are two kinds of perturbation tests: positive and negative. In the positive perturbation test, the tokens are masked from most to least relevant, and a sharp performance drop is expected when significant pixels are masked out. In contrast, negative perturbation tests involve masking tokens from least to most relevant, and a successful explanation would preserve model accuracy while removing irrelevant pixels (Chefer et al., 2021a). This test implies that the model remains insensitive to unimportant pixels while remaining sensitive to crucial ones (Böhle et al., 2023).

- **Pointing Game** is a method for evaluating the saliency maps of the explanation in comparison to human-annotated bounding boxes. These bounding boxes identify the parts of the input data that humans consider essential for the model's prediction. If the pixel with the highest saliency value falls within the human-annotated bounding box, it is counted as a hit (#hits). Otherwise, it is counted as a miss (#misses). The Pointing Game accuracy is defined as follows:

$$\texttt{accuracy} = \frac{\#\texttt{hits}}{\#\texttt{hits} + \#\text{misses}} \tag{12}$$

This metric can be employed to evaluate the weighted average of classification accuracy and explainability, meaning that it considers both the correctness of the prediction and the clarity of the explanation (Zhang et al., 2018). Pointing Game has also been extended to be applied to the image grids. **Grid Pointing Game** involves generating a series of synthetic image grids, each containing multiple unique objects. The goal of this test is to explain a given class by measuring the fraction of positive attribution that an explanation method assigns to the correct sub-image (grid) (Bohle et al., 2021).

- **Segmentation Tests** consider each visualization as a soft segmentation of the image and compare them to the ground truth provided in the dataset. Explanations are generated based on the predicted class and assessed against the ground truth using four metrics: pixel accuracy, mean IOU (mIoU), mean Average Precision (mAP), and mean F1 (mF1). Pixel accuracy and mIoU are calculated by binarizing the explanation using a threshold determined by the average of the attribution scores. On the other hand, mAP and mF1 are calculated by averaging the corresponding scores across multiple threshold levels (Chen et al., 2023).

- **Sensitivity-n** has been proposed to test specific attribution values rather than considering only the importance rankings. It does this by measuring how much the model's prediction changes when a feature is removed. This correlation is calculated for many different subsets of features and then averaged. Sensitivity-n works by randomly selecting subsets of features and calculating the correlation between the model's prediction with those features and the sum of the attribution scores for those features. A high sensitivity-n score indicates that the attributions are accurate and correlate well with the impact on the model's prediction when a feature is removed (Covert et al., 2023; Ancona et al., 2017).

# 5 Tools and frameworks

Different explainability techniques have been developed to provide insight into deep learning models and facilitate comprehension of their decision-making processes. Implementing these techniques from scratch can be a significant challenge. Fortunately, there are numerous libraries specifically designed to aid in this task. This section will introduce some of these libraries and explore their features and capabilities.

• **Captum** is a unified and open-source toolkit for PyTorch. This library has been particularly designed for the interpretability of ML models. It provides implementations of various gradient and perturbation-based attribution algorithms, which can be widely used at the level of features, neurons, or layers. Furthermore, the Captum library has a robust set of evaluation metrics and is highly extensible, versatile, and applicable to classification and non-classification models. One of the key advantages of this library is its ability to support diverse input types, including images, texts, audio, and videos (Kokhlikyan et al., 2020). Thereby, by providing a comprehensive suite of tools and making them widely accessible, Captum empowers PyTorch users to not only better understand the inner workings of the models but also to diagnose and troubleshoot issues as they arise.

• **InterpretDL** is a model interpretation toolkit providing a collection of classical and new interpretation algorithms. With this toolkit, researchers have access to a variety of algorithms such as LIME, Grad-CAM, and Integrated Gradients for explaining CNNs, multi-layer perceptrons, transformers, and ViTs. Notably, even researchers working on developing new interpretation algorithms can benefit from this toolkit, as it provides a benchmark for comparing their work with the existing ones. This toolkit streamlines the model interpretation process, helping developers to effectively analyze the performance of their models and make improvements where necessary (Li et al., 2022).

• **AttentionViz** offers a global view of the transformer's attention. It is a particularly designed tool for visualizing attention patterns across both language and vision transformers. This visualization tool provides a special insight into query-key embedding interactions at different layers, which enables users to analyze global patterns across multiple input sequences. AttentionViz helps users explore attention patterns at scale, making it a useful platform for researchers working with transformers (Yeh et al., 2023).

• **Quantus** is a cutting-edge toolkit designed to facilitate the evaluation of neural network explanations. Before the development of Quantus, there was a marked absence of specialized tools for evaluating the reliability and accuracy of the explanations. This gap in the field resulted in the development of Quantus, which is a powerful Python-based evaluation toolkit with a well-organized and vast collection of evaluation metrics and tutorials tailored for the assessment of explainable methods. This toolkit is offered under an open-source license on PyPi (Hedström et al., 2023).

# 6 Datasets

In the aforementioned reviewed methods, various datasets including CIFAR, ImageNet, and COVID-QU-Ex were employed. CIFAR-10 is a commonly used dataset consisting of $60,000$ small, colored images categorized into 10 classes. This dataset is frequently used to train and evaluate image classification algorithms (Krizhevsky & Hinton, 2009). ImageNet is also a well-known massive dataset comprising millions of labeled images across thousands of classes. ILSVRC-12, particularly the 2012 dataset, serves as a substantial resource for training and evaluating different CV tasks (Deng et al., 2009). Regarding COVID-QU-Ex, it is a medical dataset comprising $33,920$ chest X-ray images that aid in the diagnosis of COVID-19, healthy cases, and other non-COVID infections such as pneumonia (Tahir et al., 2021).

For the other CV tasks, the ImageNet-Segmentation dataset (Guillaumin et al., 2014) was adapted for the segmentation-based experiments. In addition, Pan et al. (2021a) in IA-RED$^2$ employed the Kinetics-400 dataset for video classification purposes (Carreira & Zisserman, 2017).

# 7 Experiments and results

In this section, some of the aforementioned methods specifically introduced for the explainability of vision transformers are examined. These techniques include attention-based methods, which attempt to discover significant aspects of data based on the attention mechanism, and pruning-based approaches, which streamline the network architecture by removing unnecessary components. By examining these methodologies, we gain invaluable insights into their strengths and limitations concerning explainability metrics, which contributes to the pursuit of transparent and interpretable AI systems.

## 7.1 Explanation evaluation of the attention-based methods

In this section, the attention-based methods are evaluated using the Deletion and Insertion measures introduced in section 4. To do so, we selected six diverse techniques including Rollout, PartialLRP, Grad-SAM, Beyond Attention, TAM, and Beyond Intuition, all of which have been fully described in section 3.2. The rationale for selecting these techniques lies in their diverse approaches for generating heatmaps for explaining model predictions, encompassing gradient-based methods to more advanced strategies.

Throughout the experiments, 100 classes were randomly selected from the ImageNet validation set. They comprised a total of $5,000$ images. Each image was passed through each of the attention-based techniques, and their resulting heatmaps were utilized for Insertion and Deletion calculations. In the deletion process, the most significant pixels within the heatmaps were progressively masked and the target class probabilities were recorded. Conversely, for insertion evaluation, the most important pixels from the generated heatmaps were gradually inserted into a blank image, while the probability of the target class was captured at each step. Fig. 6 shows the trend of probability changes for these processes, where in Deletion, the probability of the target class decreases and in Insertion, the probability increases, as expected.

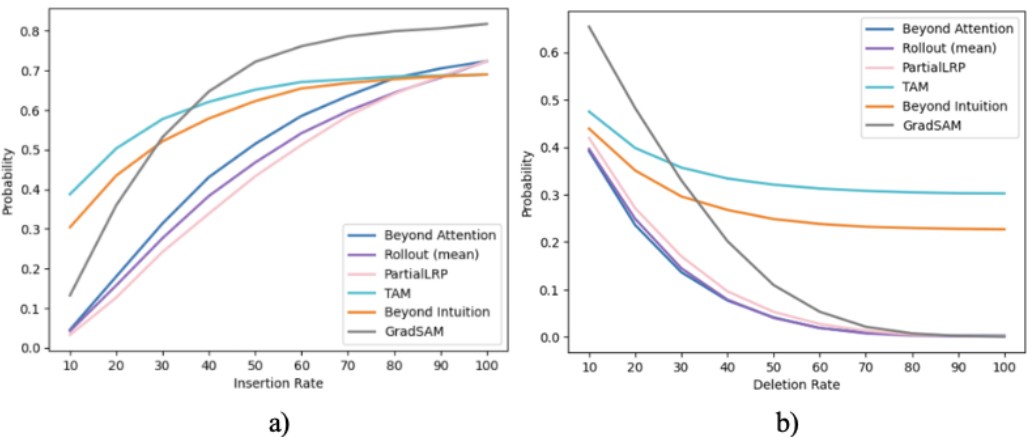

Figure 6: Visualization of the impact of a) inserting and b) deleting the most significant input pixels, indicated by different attention-based explainability techniques, on the probability of the target classes.

The Insertion and Deletion AUCs are presented in Table 2. As described in section 4, lower Deletion scores and higher Insertion scores denote better performance. Our findings indicate that the Beyond Attention method, a synergy of LRP and Rollout techniques, outperforms the original PartialLRP and Rollout approaches. In addition, Beyond Attention is the well-performing method in the case of Deletion and GradSam in the case of Insertion. Nevertheless, no singular method emerged as the best in both Deletion and Insertion criteria, emphasizing the importance of quantitative measures rather than employing only quantitative evaluations in developing attention-based explainable techniques. This necessitates specific continued research in the intersection of XAI methods and novel network architectures such as ViTs.

Table 2: The Deletion and Insertion AUCs of the various attention-based explainability methods

|  | Rollout(mean) | PartialLRP | GradSAM | TAM | Beyond Attention | Beyond Intuition |
|---|---|---|---|---|---|---|
| Deletion ($\downarrow$) | 0.074 | 0.08 | 0.15 | 0.30 | **0.072** | 0.24 |
| Insertion ($\uparrow$) | 0.41 | 0.39 | **0.58** | 0.56 | 0.44 | 0.53 |

## 7.2 Explanation evaluation of the pruning-based methods

This section presents evaluations conducted on pruning-based methods that indirectly impact the explainability of vision transformers. As mentioned in section 3.4, these methods primarily aimed to enhance the efficiency of ViT by retaining only informative image patches, which later claimed to have a positive effect on the models' explainability as well. The rationale behind these experiments is that in the context of pruning-based methods, the group of explicitly explainable ones utilized appropriate evaluation metrics to demonstrate their performance in the explainability perspectives. However, the implicitly explainable methods have solely proved their claim of having benefits on explainability through visualization. Therefore, our focus on implicitly explainable methods aims to assess the validity of these claims by providing a more comprehensive quantitative evaluation, thereby enhancing the understanding of their contribution to model explainability.

Hence, to bridge this gap, we employed two well-known implicitly explainable methods, namely Dynamic ViT and EViT, for evaluation and comparison of this section. Both of these methods have the same strategy and identify the most crucial patches of the images to retain. Specifically, Dynamic ViT directly removes uninformative patches, while EViT fuses the less relevant ones into single units. The basis of our subsequent analysis lies in assessing the models' accuracy as various proportions of an image's patches are pruned.

To conduct consistent comparisons, both Dynamic ViT and EViT models were employed with the DeiT-S backbone (Touvron et al., 2021a). Additionally, a subset of $8,000$ images from $160$ random classes of the ImageNet validation set were used to compute the measures. These two models predict image classes by varying the `Keep_Rate`, which determines the proportions of retained patches in the input images.

The experiments were conducted with three different `Keep_Rates`, which involved three stages of reducing number of image patches. Therefore, the models generated three predictions for every image. This allowed us to observe changes in their ability to accurately identify the proper class when they relied on less information. Indeed, the average of the predicted probabilities across all the classes provides a metric for assessing how effectively each model maintains its discrimination capability despite processing fewer image patches.

Fig 7 shows the results of the experiments. As expected, the lower `Keep_Rates` leads to lower prediction probability for both methods. This decline occurs because, with fewer tokens, the models have limited information to predict. However, Dynamic ViT descends more gradually compared to EViT, resulting in higher and better AUC. This indicates that Dynamic ViT is more adept at identifying and removing truly uninformative patches that have minimal influence on predicting the correct classes. Consequently, Dynamic ViT offers better explainability by focusing on processing only the most relevant image sections associated with the main objects.

To provide a clearer understanding of the impact of pruning on the explainability perspective, we present Fig 8. This figure visualizes the results of the models applied to three images belonging to various classes. The models are expected to remove unnecessary image patches. As the `Keep_Rate` decreases, EViT progressively removes larger portions of the image, including potentially informative regions for class prediction. This observation holds true for all three images, where the main objects (e.g., bird, shark) are predominantly removed during the third stage of the pruning (`Keep_Rate` $= 0.3$ ). In contrast, Dynamic ViT consistently preserves the primary objects while masking the background patches. This observation implies that Dynamic ViT offers a more explainable approach to token pruning, which is in accordance with the previously mentioned quantitative results.

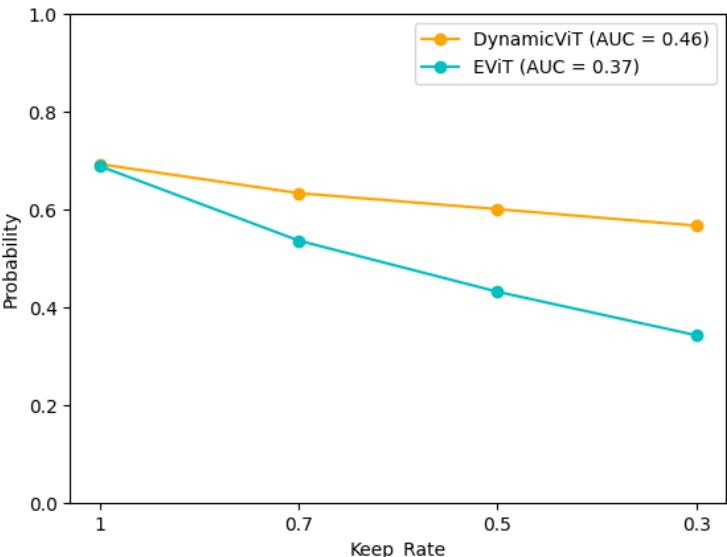

Figure 7: This figure shows how prediction accuracy drops for Dynamic ViT and EViT as the number of image patches, which are processed (`Keep_Rate`), decreases. Dynamic ViT maintains higher prediction probability with lower `Keep_Rates` compared to EViT, hinting at its potential for better explainability by focusing on truly informative data while pruning uninformative patches.

## 8   Discussion and future works

We have comprehensively reviewed the existing explainability techniques for ViT, grouping them into five categories, and delving into the cutting-edge research of each category. In terms of attribution methods, most of the model-agnostic methods designed for the DNNs are also applicable to the transformers. However, leveraging the attention weights can enhance the interpretability of these methods given the knowledge gained by the attention mechanism. In this context, feature attribution has received more attention than feature visualization, which can be rectified in future studies to further elucidate the global model's working procedure (Ghiasi et al., 2022).

Direct utilization of attention weights has been observed to fall short in vision tasks, raising concerns about the suitability of attention for explanations (Chefer et al., 2021a). They should be viewed as only one part of a complex series of non-linear operations that may not be sufficient to comprehensively understand different dependencies within a model. Hence, there is a pressing need for robust discussion in research on the appropriateness of attention weights for interpretation purposes (Jain & Wallace, 2019). Additionally, the methodology of combining attention matrices in the multi-head self-attention blocks requires further scrutiny to provide robust explanations (Voita et al., 2019), and rather than relying solely on linear combinations of the final attentions of the model's layers, exploring different mathematical methods could yield better results.

When using pruning-based methods, it is crucial to pay more attention to the explainability aspect in order to fully utilize the resulting non-dense and more transparent architectures. Additionally, most of these methods, even the explicitly explainable ones have not been evaluated using the explainability quantitative criteria to ensure their faithfulness and completeness. This underscores the need for further research into the explainability of pruning-based methods and a more rigorous evaluation of their explainability.

The proposed taxonomy was specifically provided for vision transformers and did not necessarily follow the traditional borders like scope, applicability, input or output formats, and so on, which mostly apply to categorize the explainability methods of the ML methods, in general, (Arrieta et al., 2020). However, each of these perspectives can be seen in addition to our perspective. For example, according to the developing methodology which divides the methods into two main categories of post-hoc and ad-hoc methods, most of

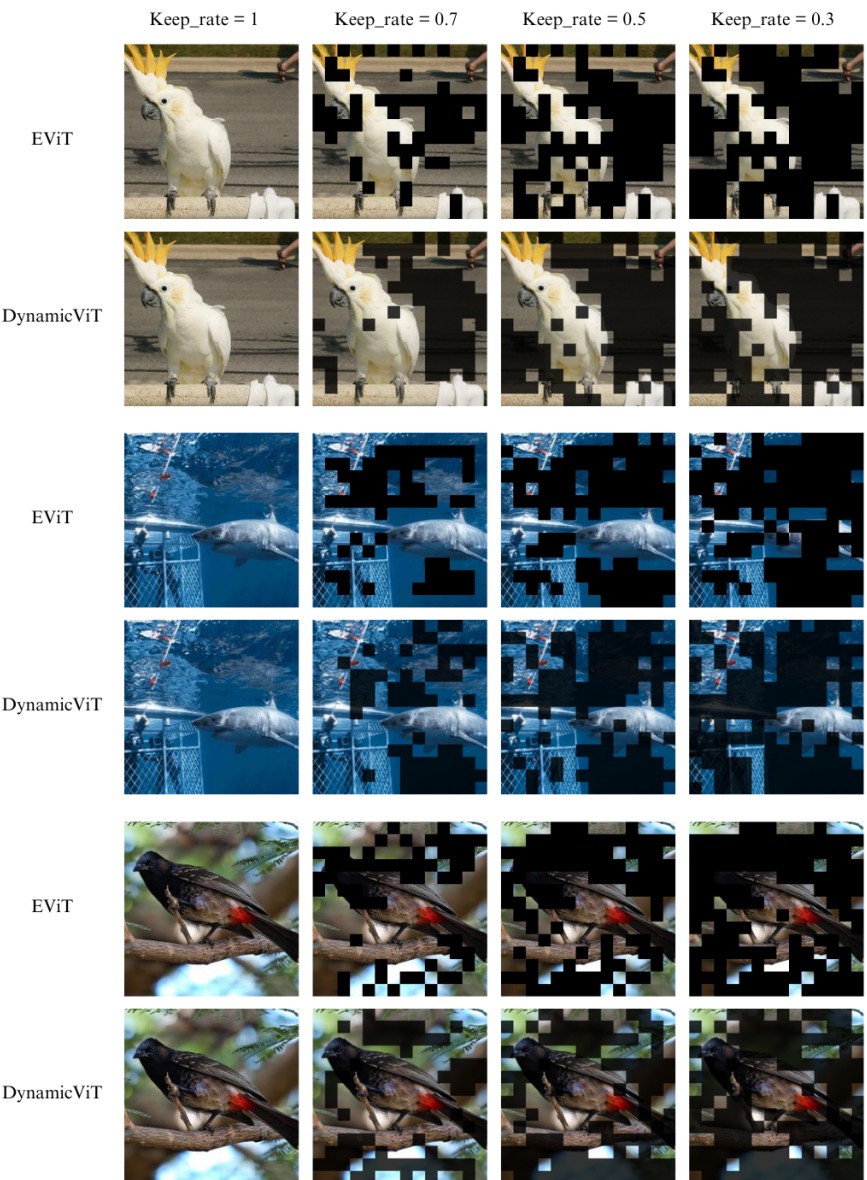

Figure 8: This figure showcases the impact of Dynamic ViT and EViT on the three images of the ImageNet belonging to the Cockatoo, Shark, and Bulbul classes, respectively. The visualizations demonstrate the relationship between `Keep_Rates` and the performance of each method. The masked patches indicate which tokens are removed in Dynamic ViT or fused in EViT. Dynamic ViT seems to prune tokens in a more interpretable manner. It preserves patches that humans typically utilize for image recognition while EViT unexpectedly discards some of these potentially informative tokens.

the methods of the first two categories, namely attribution and attention-based methods are post-hoc, and most of the inherently explainable methods are ad-hoc.

Different inherently methods discussed in this paper follow different working procedures. In the area of inherently explainable methods, the trade-off between explainability and performance is often less discussed compared to interpretable models. This is because these models are designed to provide insights into their decision-making process, making it easier to understand how they arrive at their predictions and the com-

putational issues are not usually a matter of concern. However, there are still some trade-offs to consider in developing such models.

Inherently explainable methods might require additional computational resources to compute the explanations, which can slow down the model's inference time. These methods often require more complex architectures or additional components to generate explanations. This can lead to a higher risk of overfitting or decreased performance. In addition, the choice of hyperparameters for inherently explainable models can be more nuanced, as they need to balance the trade-off between performance and interpretability. Finally, they may require additional data or specific data preprocessing to generate high-quality explanations.

We acknowledge that while most existing research on the explainability of ViT has focused on classification tasks, there is a necessity to explore their potential in other tasks. Future work should separately investigate the explainability of ViT in tasks like segmentation, object detection, and visual question answering, which could provide valuable insights and enhance the application of these models.

A new area of XAI research focuses on Generative AI (GenAI) and multi-modal applications. With the emergence of Generative AI (GenAI), there is a critical need for explainable mechanisms to comprehend and trust the outputs of these systems. Explanations enhance the verifiability of generated content, thus addressing major issues like hallucinations. However, explaining GenAI is particularly challenging due to its inherent complexity. The stochastic nature of these models results in high variability and unpredictability in their outputs. Additionally, the multi-modality aspect of GenAI, which integrates various types of data, further complicates the task of providing consistent and understandable explanations. In this context, XAI techniques are structured based on novel dimensions such as modality, dynamics, foundational source, and required access, enhancing the adaptability and clarity of AI model explanations for diverse user needs. Moreover, evaluating GenAI explanations is challenging because traditional metrics may not capture the full picture, necessitating new evaluation methods (Schneider, 2024). Given these complexities, exploring explanations of these applications requires further effort.

Finally, it is worth noting that there has been a notable lack of research on how to utilize the benefits of explainability methods to facilitate model debugging and improvement as well as increase model fairness and reliability, especially in the ViT applications. However, by leveraging these insights, it is conceivable to instill greater trustworthiness and fairness into ViT models, resulting in improved decision-making and ultimately, a higher success rate in real-world applications.

## 9 Conclusion

In this paper, we presented a comprehensive overview of explainability techniques proposed for vision transformers. We provided a taxonomy of the methods based on their motivations, structures, and application scenarios and categorized them into five groups. In addition, we detailed the explainability evaluation criteria as well as explainability tools and frameworks. Finally, several essential but unexplored issues to enhance the explainability of visual transformers were discussed, and potential research directions were further suggested for future investment. We do expect that this review paper can help readers obtain a better understanding of the inner mechanisms of vision transformers as well as highlight open problems for future work.

## Acknowledgments

The authors would like to thank the anonymous reviewers for their comments and suggestions that helped us to improve the quality of this paper.

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
