# OpenReview forum: "Explainability of Vision Transformers: A Comprehensive Review and New Perspectives"
_TMLR — Rejected by TMLR_

### Review · Reviewer_SkfT · 2024-06-19

**Summary Of Contributions:**

This survey paper reviews techniques, evaluation criteria, software packages, and datasets in explainability of vision transformers. The techniques are organized into common attribution, attention-based, pruning-based, inherently explainable, and applicable to non-classification tasks. Two experiments and a visualization are also provided to compare some of the reviewed techniques. The paper concludes by highlighting current limitations of explainability research in terms of techniques, evaluations, and applications.

**Audience:**

Yes

**Claims And Evidence:**

Yes

**Requested Changes:**

See the comments on writing.

**Strengths And Weaknesses:**

Although I am not well-versed enough in explainability research to ensure the paper has not missed a key method, the review appears comprehensive. The taxonomy in Figure 2 is especially helpful for research in this area. The techniques, evaluation criteria, datasets, tools, and limitations of current reserach are all adequately explained. I am overall positive about the submission as long as the following comments about writing are addressed and unless other issues are brought up in the reviewer discussions.

1. Section 5 has a footnote that PaddlePaddle is "The first independent R&D deep learning platform in China". This is irrelevant to the paper and should be removed. Further, in case there is some conflict of interest regarding the software packages, the authors should also consult the journal's guidelines on conflicts of interest and sources of funding to ensure the final version is transparent.

2. The text in Section 5 appears AI-generated. I cannot guarantee this and I cannot find the TMLR guidelines on AI-generated text if this is the case. Depending on the action editor's decision, this section may have to be paraphrased.

3. The same section has multiple vague phrases when introducing the tools such as "a valuable resource", "a noteworthy ... tool", "a unique perspective", and an "invaluable resource". Even if the text is not AI-generated, such phrases should be removed.

4. Many of the citations throughout the paper should have parenthesis using \pcite or \parencite.

5. Section 3 has multiple comments about the performance of the methods. Since this section appears before the evaluation criteria is described, it is important to mention, at least after the first comment on performance, that the next section will describe the evaluation criteria.

6. The description for Attention Rollout says "Although it outperforms the raw attention explanations, it has several limitations." The name of the evaluation criteria and at least one of the limitations should be mentioned. This is also the best place to refer the reader to section 4 for a description of the evaluation criteria as per comment 5 above.  Similarly, the text about Beyond Intuition mentions that it "performs 20 times more accurate than ...". The name of the criterion should be mentioned. Also, accurate->accurately.

7. Attention Flow is said to be "more correlated than the Rollout method in some cases". What quantity is it correlated with?

8. The text below Eq. 12 mentions that this metric is "a weighted average of classification accuracy and explainability". Eq. 12 does not look like a weighted average. Does this text refer to Eq. 12 itself, or is Eq. 12 used in some weighted average to obtain another quantity?

9. There are some phrases in the text that will be confusing to some readers and will need a clearer definition, an example, or rephrasing. Important cases are "face information retention" (Section 2.1), "high-level semantics" (Section 3.2), and "class token" and "image token" (Section 3.3.2).

10. Is it possible to use the same x-axis for Figure 6 (a) and Figure 7 instead of insertion rate and keep_rate?

11. Typos and minor errors: "Shaply"->"Shapley" (Section 3.1), "qualitative and qualitative"->"qualitative and quantitative" (Section 3.3.2). "extended Beyond Attention method"->"proposed Beyond Attention method".

---

> ### Author Response · Authors · 2024-07-22
> **Reply to  Reviewer SkfT**
>
> First of all, We thank the reviewer very much for his/her positive comments on our submitted paper. We have done our best to improve the article by considering the following numbered list.
>
> 1: We followed your suggestion and removed the footnote. Given that PaddlePaddle and InterpretDL have common contributors, we concluded that there is no conflict of interest and therefore we removed that.
>
> 2: We guarantee that this section ( section 5 ) is not AI-generated, however, we paraphrased some parts that may seem more formal than necessary.
> 3: Given the above comments, we revised that section and removed/ changed those phrases.
>
> 4: Thanks for this comment, we refined all citations according to the TMLR template.
>
> 5: We followed your suggestion and included a note to Section 3 (end of 4th paragraph), indicating that Section 4 will provide a detailed description of the evaluation criteria.
>
> 6: Thanks for this comment, We added the names of the evaluation metrics and the limitations to the descriptions. Additionally, we fixed the typos in these paragraphs (4th paragraph in section 3.2):
>
> Although it outperforms the raw attention explanations, it has several limitations. One of these is the exaggerated differences between attention scores for different tokens, which occur because it combines tokens linearly through layers based on attention weights.  This results in a loss of faithfulness in saliency maps. Faithfulness refers to how accurately a saliency map represents the target model. It is measured by observing how quickly the model’s score for the target class changes when important pixels are removed or added. This loss of faithfulness affects evaluation metrics such as Deletion Area Under the Curve (AUC) and Insertion AUC (for a detailed description of these criteria please see Section 4 ) (Komorowski et al., 2023; Bhatt et al., 2020; Petsiuk et al., 2018; Nguyen & Martínez, 2020).
>
> and in the 11th paragraph in section 3.2 as follows:
>
> Specifically, the token-wise attention map performs 20 times more accurately than Attention Rollout in terms of faithfulness evaluation metrics, such as Deletion AUC, Insertion AUC, and Faithfulness correlation (see section 4) (Komorowski et al., 2023; Bhatt et al., 2020; Petsiuk et al., 2018; Nguyen & Martínez, 2020)
>
> 7:  We added the quantity of correlation as follows:
>
> It is known to be more correlated with the importance scores of the input tokens than the Rollout method with respect to the ablation method and input gradients, but it is too slow to support large-scale evaluations (Abnar & Zuidema, 2020).
>
> 8: This metric evaluates explainability, considering the clarity of the explanation, with overall performance measured by the mean accuracy across different categories in the dataset. In addition, it can be employed to evaluate the weighted average of classification accuracy and explainability, meaning that it considers both the correctness of the prediction and the clarity of the explanation. As these statements, we revised that section as follows:
>
> This metric can be employed to evaluate the weighted average of classification accuracy and explainability, meaning that it considers both the correctness of the prediction and the clarity of the explanation (Zhang et al., 2018).

---

> > ### Author Response · Authors · 2024-07-22
> > **Reply to Reviewer SkfT _ part 2**
> >
> > 9: We addressed the point you raised  and clarified those phrases as follows:
> >
> > Face information retention refers to the challenge of maintaining information across lengthy sequences in traditional encoder-decoder models. However, due to the vagueness, we revised the corresponding paragraph  (Section 2.1) as follows:
> >
> > Transformers revolutionized machine translation by overcoming the constraints of sequential transduction models. Those models rely on an encoder-decoder setup with a fixed-size context vector and struggle to maintain information over long sequences. Indeed, Transformers eliminated recurrent neural networks within blocks, improving parallel processing and mitigating challenges like vanishing and exploding gradients (Vaswani et al., 2017; Sutskever et al., 2014).
> >
> >
> > High-level semantics means the richer representations of the input data that the model perceives as it processes information through the layers using mechanisms like attention. We revised the corresponding paragraph ( 10th paragraph in Section 3.2) as follows:
> >
> > This method propagates information from top to bottom. It computes the relevance between high-level semantics and input features. High-level semantics are used to represent richer patterns and concepts and are processed through the attention mechanism in layers. The input features include edges, textures, and basic shapes.
> >
> > Class Token and Image Token: class token is an extra token [CLS] that is added to the set of image tokens and is responsible for aggregating global image information and final classification, while image tokens refer to the image patches, which are treated as individual units. This clarification was added to the manuscript section 3.3.2 as follows:
> >
> > The class token is an extra token (CLS) added to the set of image tokens and is responsible for aggregating global image information and final classification. In contrast, image tokens refer to the image patches, which are treated as individual units.
> >
> > 10: In fact, the deletion rate and keep rate refer to similar concepts, both of which evaluate the performance of the model when some components of the input are removed (deletion rate) or in other words, the remaining components are preserved (keep rate). In Figure 6, the components are pixels, while in Figure 7, they are image patches. The former is typically used to assess model performance from an explanatory perspective, whereas the latter is used from an efficiency point of view. We report our results using these terms for two reasons: to preserve the original concepts used in the corresponding papers and to differentiate between the employed contexts.
> >
> >
> > 11: Thank you for this comment. Regarding your mention and our careful review, all these kinds of typos and minor errors were corrected.

---

> > > ### Comment · Reviewer_SkfT · 2024-08-16
> > >
> > > Thank you for the detailed response. The revisions addresses all my comments.

---

### Review · Reviewer_2moi · 2024-07-02

**Summary Of Contributions:**

This paper organize and summarize works in explainability of Vision Transformers and categorize the existing work into common attribution approaches, attention-based methods, pruning based methods, inherently explainable methods and others. The paper first provide detailed introduction to the mechanism of vit. Then, it provides discussions of the representative work for each of the aforementioned categories. Finally, some of the methods are empirically evaluated and compared.

**Audience:**

Yes

**Broader Impact Concerns:**

The broader impact are sufficiently addressed in the discussion and future works section, and conclusion section.

**Claims And Evidence:**

Yes

**Requested Changes:**

Similar to weakness:

- Please add more discussions in post-hoc and ad-hoc categories.
- More detailed discussion on intrinsic explainable/ interpretable methods are extremely encouraged.
- Please add more explicit discussions on how those 'implicit methods' could be related to model explainability.
- Empirical evaluations on explicit explainable methods are much more preferred than that of those so-called 'implicit methods'
- The figure of Taxonomy of explainability methods for Vision Transformers should clearly indicate the type of tasks models are addressing such as classification, segmentation and etc.

**Strengths And Weaknesses:**

Strength:
- The structure of the paper is very clean. It is easy to follow and understand the content in each of the sections.
- The work presented in this paper is model-wise comprehensive and the detailed introduction to ViT is helpful for readers to recall and understand the topic.
- The empirical evaluation and its result provide some interesting insights.

Weakness:
- Most of work presented in this paper belong to post-hoc methodologies such as those in attribution based methods, and attentions based methods. These are not explicitly discussed and introduced in the paper. The paper should first divide works into post-hoc and ad-hoc explainability methods, and then divide those into sub-categories such as attention-based methods and etc. Otherwise, the work could be misleading.
-  The paper also include some work that are not directly addressing the explainability of the models such as the implicit explainable methods and possibly explainable methods. Though works in implicit explainable methods could provide some insights on vit mechanism, those work are mainly designed for model efficiency. Whether those works are explainable is controversial, and explicitly include those into a survey paper as one of the main section and one of main part of empirical evaluation in the experiment section are misleading.
- The paper is lack of more detailed discussion in intrinsic explainable models, and empirical evaluations on those.
- Once again, when empirically evaluate pruning-based methods, only two works mentioned in the section of implicit explainable methods are examined. The results seem to have weak connections to the topic -- model explainability.
- The figure of Taxonomy of explainability methods for Vision Transformers should clearly indicate the type of tasks models are addressing such as classification, segmentation and etc.

---

> ### Author Response · Authors · 2024-07-25
> **Reply to 2moi**
>
> First of all, We thank the reviewer very much for his/her positive comments on our submitted paper. We have done our best to improve the article by considering the following numbered list.
>
> 1. The proposed taxonomy looks at the Vit-based methods from the window of explainability and categorizes them based on their motivations, structures, and application scenarios. We cover a wide range of novel studies rather than attribution and attention-based ones. The proposed taxonomy is specifically provided for vision transformer and does not necessarily follow the traditional borders like scope (local/global), applicability (model-based/agnostic), input or output formats, and so on, which mostly apply to categorize the explainability methods of the ML methods in general (fig 6 in (Arrieta et al, 2020) ). However, each of these perspectives can be seen in addition to our perspective. For example, according to the developing methodology which divides the methods into two main categories of post-hoc and ad-hoc methods, most of the methods of the first two categories, namely attribution and attention-based methods are post-hoc, and most of the inherently explainable methods are ad-hoc. Among the pruning-based ones, some of them (X-Pruner, Vision DiffMask, Patch Slimming) are post-hoc and some (Dynamic ViT, EViT, IA-RED\textsuperscript{2}, HVT) are ad-hoc.  Regarding the last category, the same story is true and some methods are post-hoc-and some others are ad-hoc. Therefore, since there is no strict order between the methods, it is not possible to first divide works into post-hoc and ad-hoc explainability methods, and then divide them into sub-categories.
>
>
> Given this comment, we added the following description to the 5th paragraph of section 8:
>
> The proposed taxonomy was specifically provided for vision transformers and did not necessarily follow the traditional borders like scope, applicability, input or output formats, and so on, which mostly apply to categorize the explainability methods of the ML methods, in general (Arreita et al, 2020). However, each of these perspectives can be seen in addition to our perspective. For example, according to the developing methodology which divides the methods into two main categories of post-hoc and ad-hoc methods, most of the methods of the first two categories, namely attribution and attention-based methods are post-hoc, and most of the inherently explainable methods are ad-hoc.
>
>
>
> 2.  Given your suggestion, we added more description on intrinsic explainable and  explainable methods to the 8th and 9th paragraphs of the  introduction as follows:
>
> To advance our understanding of vision transformers, it is crucial to explore their inner working procedure and examine their explainability. Among different explainability techniques, some approaches concentrate on developing models that are able to explain themselves inherently. These self-explaining methods may attempt to take advantage of different intrinsic explainable tools such as decision trees or rule-based systems, or from another perspective, they may enforce sparsity in the model architecture. However, these models often struggle to achieve the same level of accuracy as more complex black-box models, and as a result, there must be a careful balance between explainability and performance (Rudin, 2019).
>
> In contrast with the inherently/intrinsic explainable methods, interpretable methods attempt to provide human-understandable representations of the decision-making process of the models using different strategies. These methods aim to translate complex, high-dimensional representations into a simpler, more comprehensible format. Explainability methods as debugging tools, gaining insights into the limitations, unintended biases, risks, and social impacts of the models. This enables us to develop reliable, ethical, and safe models that can effectively be deployed in real-world scenarios. Additionally, they facilitate tracking of the models' capabilities over time, compare them with other models, and consequently enhance the model's performance (Ali et al, 2023).
>
> and in section 3.4 as follows:
>
> These explainable methods aim to understand how the model's working procedure, such as the weights and biases of a neural network, contribute to its output. In the context of DNN, they attempt to develop models that are able to explain themselves inherently. They may take advantage of different intrinsic interpretable tools as well.

---

> ### Author Response · Authors · 2024-07-25
> **Reply to 2moi_Part 2**
>
> 3. Regarding the explicit and implicit explainable methods, the former methods focus on pruning some model's components specifically to enhance the explainability of vision transformers. On the other hand, the other methods apply pruning techniques to increase efficiency, however, they have been shown to positively impact model explainability as well. These methods do not explicitly prioritize explainability as their primary goal and this is why we call them implicitly explainable methods. Indeed, the explainability improvement is a beneficial byproduct of their design. Therefore, based on your comment, we clarified the differences of these methods in the first paragraph of Section 3.3.
>
> 4. How those 'implicit methods' could be related to model explainability?
>
> Implicitly explainable methods are initially designed to improve model efficiency, but they have clearly demonstrated through visualization that their pruning approaches would result in positive effects on the models’ explainability as well (Rao et al, 2021: Liang et al, 2021). Indeed, while their main goal is to improve efficiency, the clarity that comes from simplifying the model also provides valuable insights into how the model works and makes decisions. This dual benefit highlights the importance of implicitly explainable methods in enhancing both the efficiency and explainability of vision transformers. Therefore, the characterization of these methods as explainable is substantiated by the evidence presented in the cited references. However, their qualitative performance needs further evaluation, which we have focused on in the paper. We addressed the point you raised and clarified these statements in the first paragraph of Section 3.3.2.
>
> 5. Why Empirical evaluation is not on explicit explainable and is on implicit?
>
> We have chosen to evaluate implicitly explainable methods because explicitly explainable pruning methods, as in (Pan et al, 2021a; Yu &Xiang, 2023: Nalmpantis et al, 2023), already utilized well-established evaluation metrics to demonstrate their performance in terms of explainability. These methods have employed evaluation curves similar to those used in our experiments to examine how accuracy drops with different pruning rates. This metric is reliable because when the accuracy drops gradually (higher AUC), it indicates that the method is more adept at identifying and removing truly uninformative patches that have minimal influence on predicting the correct classes, therefore offering better explainability. In contrast, implicitly explainable methods claim to have positive impacts on explainability and have only showcased their capabilities through visualizations in the referenced papers. Therefore, our focus on implicitly explainable methods aims to assess the validity of these claims by providing a more comprehensive quantitative evaluation, thereby enhancing the understanding of their contribution to model explainability. We also plot evaluation curves in Figure 7 of our paper for two implicitly explainable methods to observe how the average of the predicted probabilities across all the classes drops with different keep rates, to analyze changes in their abilities to accurately identify the correct class when relying on less information. Furthermore, we compare the performance of two implicitly explainable methods that share similarities in their pruning approaches. Nevertheless, we acknowledge the need for more experiments in this area to provide a broader perspective. As these statements, we revised the first paragraph of Section 7.2 as follows:
>
> This section presents evaluations conducted on pruning-based methods that indirectly impact the explainability of vision transformers. As mentioned in section 3.3, these methods primarily aimed to enhance the efficiency of ViT by retaining only informative image patches, which were later claimed to have a positive effect on the models' explainability as well. The rationale behind these experiments is that in the context of pruning-based methods, the group of explicitly explainable ones utilized appropriate evaluation metrics to demonstrate their performance in the explainability perspectives. However, the implicitly explainable methods have solely proved their claim of having benefits on explainability through visualization. Therefore, our focus on implicitly explainable methods aims to assess the validity of these claims by providing a more comprehensive quantitative evaluation, thereby enhancing the understanding of their contribution to model explainability.
>
> Hence, to bridge this gap, we employed two well-known implicitly explainable methods, namely Dynamic ViT and EViT, for evaluation and comparison of this section...
>
> 6. Refine taxonomy
> Thanks for your suggestion. We will indicate the type of tasks models are addressing in Figure 2 as: classification(C),  segmentation (S), …, and  for those not specified, it is Classification default task.

---

> > ### Comment · Reviewer_2moi · 2024-07-29
> >
> > Thank you for the updates and response. I am happy with the modifications.

---

### Review · Reviewer_5rCB · 2024-07-18

**Summary Of Contributions:**

The authors comprehensively review explainability methods for ViTs, categorizing them by motivations, structures, and applications. It details various techniques, including common attribution, attention-based, pruning-based, and inherently explainable methods, and introduces advanced methods like Attention Rollout, Attention Flow, and PartialLRP. Highlighting the balance between efficiency and interpretability, it discusses pruning methods such as IA-RED2 and X-Pruner and inherently explainable models like ViT-CX and ViT-NeT. The paper extends the application of these techniques to other computer vision tasks, reviews evaluative criteria, and presents tools and frameworks for implementation, providing valuable resources for researchers and practitioners. It offers insights into current methods, their strengths, and limitations, and suggests future research directions to enhance model transparency and trustworthiness.

**Audience:**

Yes

**Claims And Evidence:**

Yes

**Requested Changes:**

-  More detailed coverage of different tasks other than classification would be beneficial, for example, segmentation tasks in medical images.

- Expand the discussion on the explainability of multi-modal applications and vision-based generative models. This will provide a more holistic view of explainability across different types of models.

- Provide a more detailed discussion on the trade-offs between explainability and performance, particularly for inherently explainable methods. This would help readers understand the practical implications of choosing different methods.

**Strengths And Weaknesses:**

Strengths:
- Comprehensive review: The paper provides a thorough and structured review of existing explainability methods for Vision Transformers (ViTs), covering a wide range of techniques and categorizing them effectively.

- Detailed method descriptions: The paper includes detailed descriptions and analyses of various explainability methods, such as Attention Rollout, Attention Flow, and PartialLRP, highlighting their applications and effectiveness.

- Evaluation criteria: A comprehensive review of evaluative criteria for explainability methods is provided, which helps in assessing the effectiveness and reliability of different techniques.

- Tools and frameworks: The paper lists and discusses various tools and frameworks available for implementing explainability methods, which is valuable for researchers and practitioners.

Weaknesses:
- While inherently explainable methods are discussed, the balance between explainability and performance could be explored in more detail, especially in terms of trade-offs not only in classification tasks.

- The explainability of multi-modal applications and vision-based generative models is briefly mentioned but not deeply explored. More detailed coverage would be beneficial, for example, segmentation tasks in medical images.

- Expanding the evaluation benchmarks to include more diverse computer vision tasks beyond classification would provide a more comprehensive assessment of the methods.

---

> ### Author Response · Authors · 2024-07-30
> **Reply to 5rCB**
>
> First of all, We thank the reviewer very much for his/her positive comments on our submitted paper. We have done our best to improve the article by considering the following numbered list.
>
> 1- Thanks for this comment. We acknowledge that expanding the evaluation benchmarks for Vision Transformers (ViT) to include more diverse computer vision tasks would provide a more comprehensive assessment of the methods. However, it is important to note that the main objective of this work is to collect and categorize the state-of-the-art fundamental explanability methods, which are mainly focused on classification tasks. Nevertheless, in response to your suggestion, we have identified two more works on the usage of explainability in Vision Transformers for medical segmentation and localization. Additionally, we added a discussion to highlight the potential of explainability of Vision Transformers in more diverse computer vision tasks in the 7th paragraph of section 8 as follows:
>
> We acknowledge that while most existing research on the explainability of ViT has focused on classification tasks, there is a necessity to explore their potential in other tasks. Future work should separately investigate the explainability of ViT in tasks like segmentation, object detection, and visual question answering, which could provide valuable insights and enhance the application of these models.
>
> Two more works, namely Ex-ViT-WBCs (Katar & Yildirim, 2023)  and TT-Net (Wang et al., 2023) were also added to section 3.4 paragraphs 6 and 7.
>
> 2- This comment is addressed in the 8th paragraph of section 8 as follows:
>
> A new area of XAI research focuses on Generative AI (GenAI) and multi-modal applications. With the emergence of Generative AI (GenAI), there is a critical need for explainable mechanisms to comprehend and trust the outputs of these systems. Explanations enhance the verifiability of generated content, thus addressing major issues like hallucinations. However, explaining GenAI is particularly challenging due to its inherent complexity. The stochastic nature of these models results in high variability and unpredictability in their outputs. Additionally, the multi-modality aspect of GenAI, which integrates various types of data, further complicates the task of providing consistent and understandable explanations.
>
> In this context, XAI techniques are structured based on novel dimensions such as modality, dynamics, foundational source, and required access, enhancing the adaptability and clarity of AI model explanations for diverse user needs. Moreover, evaluating GenAI explanations is challenging because traditional metrics may not capture the full picture, necessitating new evaluation methods (Schneider, 2024). Given these complexities, exploring explanations of these applications requires further effort.
>
>
> 3- This comment has been addressed in the 4th and 5th paragraphs of section 8 as follows:
>
> Different inherently methods discussed in this paper follow different working procedures. In the area of inherently explainable methods, the trade-off between explainability and performance is often less discussed compared to interpretable models. This is because these models are designed to provide insights into their decision-making process, making it easier to understand how they arrive at their predictions and the computational issues are not usually a matter of concern. However, there are still some trade-offs to consider in developing such models.
>
> Inherently explainable methods might require additional computational resources to compute the explanations, which can slow down the model's inference time. These methods often require more complex architectures or additional components to generate explanations. This can lead to a higher risk of overfitting or decreased performance. In addition, the choice of hyperparameters for inherently explainable models can be more nuanced, as they need to balance the trade-off between performance and interpretability. Finally, they may require additional data or specific data preprocessing to generate high-quality explanations.

---

### Comment · Action_Editor_Q9s4 · 2024-07-22
**Please address related prior work in your response**

Dear Authors,

As part of the discussion process, please also comment on the difference of your work with respect to the following two papers:

1. Chefer, Hila, et al. **Transformer Interpretability Beyond Attention Visualization.** 2021 IEEE/CVF Conference on Computer Vision and Pattern Recognition (CVPR), IEEE, 2021. Crossref, https://doi.org/10.1109/cvpr46437.2021.00084.

2. Stassin, S.; Corduant, V.; Mahmoudi, S.A.; Siebert, X. **Explainability and Evaluation of Vision Transformers: An In-Depth Experimental Study.** Electronics 2024, 13, 175. https://doi.org/10.3390/electronics13010175

Per the [TMLR guidelines](https://jmlr.org/tmlr/editorial-policies.html), surveys (reviews) must "... draw new connections, highlight trends, and suggest new problems in an area." The authors should argue that the submitted review #2757 is novel in one of these sense over papers (1) and (2), especially (2) which is also a survey of explainability in vision transformers.

Best,

AE for #2757

---

> ### Author Response · Authors · 2024-07-25
> **Reply to the AE**
>
> First of all, We thank the AE very much for his/her comment to clarify the differences between the proposed work and the available literature.
>
> The proposed work vs. the first paper:
>
> The first paper, namely "Transformer Interpretability Beyond Attention Visualization", is so-called  “Beyond Attention” in the proposed taxonomy and has been studied beforehand. This paper is a novel work on the explainability of vision transformers rather than a survey studying all the methods in this area. “Beyond Attention” introduces a new explainability technique that integrates the relevance of the attention head and its gradient with respect to the input. However, in the related work section of that paper, they compare common attribution methods such as Grad-CAM and LRP with attention-based techniques like rollout, raw attention, and partial LRP.
> In contrast, the proposed paper provides a comprehensive survey that studies the explainability techniques of vision transformers, categorizing them into five groups: common attribution methods, attention-based methods, pruning-based methods, inherently explainable techniques, and explainability of ViT in other tasks. Furthermore, the paper covers the evaluation metrics of explainability and presents experimental results as well. In addition, the proposed taxonomy includes newer works in the attention-based context, such as "TAM" and "Beyond Intuition" methods, and provides qualitative and quantitative experimental results on them, which were not covered in "the mentioned work. Additionally, our paper presents a review of explainability frameworks for vision transformers and a detailed list of datasets used in the novel methods analyzed in the paper.
>
> The proposed work vs. the second paper:
>
> The second paper, namely “Explainability and Evaluation of Vision Transformers: An In-Depth Experimental Study”, is a survey paper on the explainability of vision transformers, which is more focused on experimental and evaluation aspects of a distinct set of explainability methods. The authors consider a set of methods including several attribution ones and several attention-based methods and comprehensively compare them from different evaluative aspects. In contrast, in the proposed paper, we attempt to survey the explainability of the vision transformer from different points of view (presenting a taxonomy specifically for ViT, reviewing evaluative criteria, datasets, frameworks, new perspectives, .. ) and cover a wider range of novel studies rather than attribution and attention based ones.
>
> The main point of the proposed work is the new categorization from the explainability point of view, not only following the traditional border like scope (local/global), applicability (model-based/agnostic), and so on. As shown in Table 1 of the mentioned paper, the authors review only the common attribution and attention-based techniques. Additionally,  Figure 2 indicates that the study is limited to the aspects highlighted in blue and the authors follow the traditional taxonomy generally applied for the explainability of the ML methods. In contrast, in the proposed taxonomy, we emphasized the structural characteristics, motivations, working procedures, and application scenarios and given this matter included additional sub-sections, such as inherently explainable Vit methods and pruning-based techniques. In this regard, given the complex architecture of VIT having so many redundant units potational for applying pruning techniques, We discuss and evaluate the connection of efficiency and explainability of these methods. Furthermore, the proposed taxonomy includes the explainability of vision transformers applied to other tasks, such as segmentation, multimodal applications, and medical tasks as well.

---

### Decision · Action_Editor_Q9s4 · 2024-09-15

**Recommendation:** Reject

**Comment:**

This TMLR submission presents a review of explainability techniques for vision transformers.
The main contribution is the organization of existing literature into a taxonomy, displayed in Figure 2 of the submission.
The submission also presents experimental results in Figures 6 to 8 that briefly compare a subset of the taxonomized techniques.
Reviewers conveyed that the submission is well-written and well-organized.
These reviewers provided recommendations to accept (2 "leaning accept" and 1 "accept") this review paper, praising the coverage of interpretability and explainability work relevant for vision transformers.

Since one reviewer provided a low-quality review and did not engage during the discussion phase, I assessed the submission myself in detail.

Per the [TMLR guidelines](https://jmlr.org/tmlr/editorial-policies.html), surveys (reviews) must not only taxonomize existing work but must also "... draw new connections, highlight trends, *and* suggest new problems in an area."
In particular, surveys at TMLR must demonstrate their organization of existing work is actionable in a way that would interest researchers in the targeted area.
On this front, the authors state in the submission:

> Finally, several essential but unexplored issues to enhance the explainability of visual transformers were discussed, and potential research directions were further suggested for future investment. We do expect that this review paper can help readers obtain a better understanding of the inner mechanisms of vision transformers as well as highlight open problems for future work.

However, most of the paper is dedicated to inventorizing existing work. Suggestions for future work in Section 8, "Discussion and future work," are limited to references to existing work, in particular, challenges (e.g., challenges to interpreting attention weights) and trade-offs (e.g., "trade-off between performance and interpretability") that have been previously discussed in the literature but that are not evidenced in the survey itself.
This submission is thus light on contributions of its own beyond the collection and organization of prior work into the given taxonomy.

The authors may choose to submit a revision addressing the above concern by demonstrating that the presented taxonomy is impactful within the survey itself. An example of such a contribution is an empirical meta-analysis of relevant explainability techniques demonstrating that the presented taxonomy governs the performance or behavior of the techniques in a manner not already evident from prior works.

**Audience:**

Per the [TMLR guidelines](https://jmlr.org/tmlr/editorial-policies.html), surveys (reviews) must not only organize existing work, but must also "... draw new connections, highlight trends, *and* suggest new problems in an area." This submission presents insufficient evidence of achieving these criteria, which limits the audience.

**Claims And Evidence:**

Yes.

**Resubmission Of Major Revision:**

The authors may consider submitting a major revision at a later time.